# Spotting of Volatile Signatures through GC-MS Analysis of Bacterial and Fungal Infections in Stored Potatoes (*Solanum tuberosum* L.)

**DOI:** 10.3390/foods12102083

**Published:** 2023-05-22

**Authors:** Adinath Kate, Shikha Tiwari, Jamna Prasad Gujar, Bharat Modhera, Manoj Kumar Tripathi, Hena Ray, Alokesh Ghosh, Debabandya Mohapatra

**Affiliations:** 1ICAR—Central Institute of Agricultural Engineering, Nabibagh, Berasia Road, Bhopal 462038, India; 2Maulana Azad National Institute of Technology, Bhopal 462003, India; 3Center for Development of Advanced Computing, Kolkata 700091, India

**Keywords:** potato, storage, pectobacter infection, *Aspergillus* infection, VOC classification, multi-variate data analysis

## Abstract

Potatoes inoculated with *Pectobacterium carotovorum* spp., *Aspergillus flavus* and *Aspergillus niger*, along with healthy (control) samples, were stored at different storage temperatures (4 ± 1 °C, 8 ± 1 °C, 25 ± 1 °C) for three weeks. Volatile organic compounds (VOCs) were mapped using the headspace gas analysis through solid phase micro extraction–gas chromatography–mass spectroscopy every week. The VOC data were arranged into different groups and classified using principal component analysis (PCA) and partial least square discriminant analysis (PLS-DA) models. Based on a variable importance in projection (VIP) score > 2 and the heat map, prominent VOCs were identified as 1-butanol and 1-hexanol, which can act as biomarkers for *Pectobacter* related bacterial spoilage during storage of potatoes in different conditions. Meanwhile, hexadecanoic acid and acetic acid were signature VOCs for *A. flavus*, and hexadecane, undecane, tetracosane, octadecanoic acid, tridecene and undecene were associated with *A. niger*. The PLS-DA model performed better at classifying the VOCs of the three different species of infection and the control sample compared to PCA, with high values of R^2^ (96–99%) and Q^2^ (0.18–0.65). The model was also found to be reliable for predictability during random permutation test-based validation. This approach can be adopted for fast and accurate diagnosis of pathogenic invasion of potatoes during storage.

## 1. Introduction

The potato (*Solanum tuberosum*) is one of the important tuber crop throughout the world and is cultivated once or thrice a year, depending on the variety and climate, throughout the world. India is second in the world, producing 54.23 million tonnes in 2021 [1]. In the Indian sub-continent, potato is predominantly grown as a Rabi crop (December to March) with 90% production, followed by Kharif—6.9% (September to November) and Kharif in hills—2.8% (July to September). This commodity is stored to meet continuous market demand. Ambient storage, heap storage, pit storage and cold storage are various methods adopted for potatoes [2]. During storage, the metabolic activities of potatoes continue, resulting in various physiological changes, which can be delineated in the form of changes in chemical composition, loss of dormancy and physiological mass loss, to name a few. Storage temperature affects potato quality as well as the shelf life of potatoes to a large extent. For long-term storage, 8–12 °C is recommended, though sprouting-related losses are higher than in cold storage facilities (2–4 °C); however, the culinary properties of potatoes are compromised in refrigerated potatoes [3]. Therefore, different storage practices are followed in different regions to achieve the required quality of potatoes.

Apart from ageing-related changes, this commodity is also susceptible to various bacterial and fungal infections, which may carry over from the field or be acquired during storage and handling. The soil-borne bacteria *Pectobacterium carotovorum* causes soft rot disease, one of the most detrimental diseases. Similarly, some common postharvest diseases include black rot by *Aspergillus niger*, late blight by *Phytophthora infestans*, dry rot caused by *Fusarium* species, blight and rot by *Phytophthora capsici* and early blight by *Alternaria solani* [4]. These bacteria and fungi grow on the substrate, break down the complex compounds and emit a different set of metabolites than the uninfected samples. Some of these are volatile metabolites with a lower molecular weight and emitting odour that humans can sense even in smaller concentrations; however, some volatile organic compounds (VOCs) may remain undetected by humans until damage has significantly progressed. Various pre-harvest and post-harvest biotic and abiotic factors affect the VOC profiles of these commodities [5]. These volatile organic compounds (VOCs) can be detected through GC-MS even at a very low concentration. Identifying volatile organic compounds from uninfected and infected tubers can allow the onset detection of postharvest disease in storage [6]. FAIMs, metal-oxide-based e-sensing systems, have been developed to detect disease during the storage of potatoes [7,8,9,10]. These VOC data need to be analysed to understand the correlation between VOCs and the conditions of damage vis-a-vis normal metabolite emissions. This would help to decide which preventive measures should be taken. Several multivariate analysis techniques have been used to analyse VOC data. Principal component analysis (PCA), least discrimination analysis (LDA), artificial neural network (ANN), and partial least square discrimination analysis (PLS-DA), to name a few, are some of the widely used multivariate data analysis techniques for classifying VOC data. Khorramifar et al. [10] could classify different potato cultivars using LDA and ANN, having 100% and 96% accuracy. Sinha et al. [8] employed the quadratic discriminant analysis (QDA) technique for the development of classifiers for the classification of healthy and *Pectobacterium-carotovorum*-infected potatoes, where the QDA performed with 100% accuracy. The performance of these multivariate analytical tools largely depends on the data types.

The present work aims at envisaging weekly volatile organic compound (VOC) patterns of potatoes ageing and inoculated with *Pectobacterium carotovorum*, *Aspergillus flavus* and *Aspergillus niger* stored at various storage conditions and to classify the volatile markers using suitable multivariate data analysis techniques.

## 2. Materials and Methods

### 2.1. Microorganisms

ITCC cultures of *Pectobacterium carotovorum* spp., *Aspergillus flavus* and *Aspergillus niger* isolate were procured as slant cultures. The collected fungal cultures were grown on sterile potato dextrose agar media [11], and bacterial cultures were grown on sterile Luria agar. The bacterial suspension was prepared via adding 10 mL of 4 days old broth culture into 40 mL of sterile distilled water to form a dense suspension of around 10^8^ CFU/mL [12]. In the case of fungi, a ten-day-old culture plate was scraped out in a 10 mL sterile D/W with Tween 20 to get a thick spore suspension [13].

### 2.2. Inoculation of Potato

Uniform-size healthy and mature potatoes were brought from a local grower from Bhopal city of Madhya Pradesh, India, to the laboratory. The collected tubers were washed with running tap water, followed by distilled water and thereafter, dried under a laminar airflow hood. Potatoes were surface-sterilized with swabbing with 90% ethanol. The grid inoculation method was used for potato inoculation. A grid with four parallel lines of 2–3 mm depth orthogonal to each other was prepared, and a canter square pit was ditched, along which 50 µL of suspension was inoculated into it [13]. Control samples were also prepared with the same procedure, albeit using sterile distilled water. The inoculated and control samples were kept in airtight jars with silicon septa at different conditions: 4 ± 1 °C, 8 ± 1 °C and 25 ± 1 °C, considering the cold storage temperature and normal average ambient temperature prevailing in the northern and southern regions of India.

### 2.3. Headspace Sampling

About ½ kg of inoculated potato samples was stored in 1 L glass jars with lids on them. VOC sampling was performed through 100 µm PDMS SPME fibre (57341-U, Supelco, Milan, Italy) [6,13]. SPME fibre was kept with samples for around 35–40 min at room temperature for VOC sampling (standardised in our laboratory). Three replications of VOC sampling (VOC sampled from three jars) were performed. The experiments were conducted for 3 weeks till there was visible damage on the potato samples, as they were kept inside airtight jars without allowing outside microbiota to interfere with the sample.

### 2.4. GC-MS Analysis

The SPME fibre with absorbed VOCs was subjected to GC-MS (Model: GCMS-QP2020, Make: SHIMADZU Corporation, Tokyo, Japan) analysis through placing it for 10 min at the injection port at a temperature of 250 °C. The analysis was carried out on a Quadrupole Mass Detector system attached with an HP 5890 GC with NIST 12 and NIST 14 library arrangement, in split less mode. Helium was used as a carrier gas at a constant flow mode at a flow rate of 1 mL/min. The initial temperature of the column was kept at 60 °C for 2 min, gradually increased by a rate 5 °C/min up to 180 °C with a holding time of 2 min, and further incremented by 8 °C/min to reach a final temperature of 250 °C. The final temperature was maintained for 3 min. This protocol was standardised in our laboratory. The interface temperature was kept at 200 °C, and the ion source temperature was kept at 200 °C. The *m/z* ratio set points were from a start point of 20 to an endpoint of 500. The identification of the VOCs using GC-MS was validated via comparing the retention times with standard chemical compounds, such as acetic acid, ethylene and ethyl formate. These standard chemicals were procured from Sigma-Aldrich, St. Louis, CA, USA.

### 2.5. Multi-Variate Data Analysis

The VOCs were classified into twenty-three hyperparameters, representing the major group of compounds emitted during storage with their area percentages, as obtained from the GC-MS spectra. The VOC data represented in to four groups, i.e., non-inoculated (control), potatoes and potatoes inoculated with *Pectobacterium carotovorum* (*PB*), *Aspergillus niger* (*AN*) and *Aspergillus flavus* (*AF*), contributing up to 160 VOCs.

For the classification and analysis of data, PCA was used as an unsupervised analysis technique, which helped in reducing the mapped compounds into 2–3 major principal components via normalization, while the PLS-DA, a supervised chemometric technique [14], classified the data based on discriminant attributes of mapped VOCs [15]. The data were further analysed through hierarchical cluster analysis (HCA) performed with classification and discriminant technique using the principle of similarity measure and Ward’s method as the clustering algorithm [16]. The algorithm iteratively combines these separate clusters and forms a single cluster. The variable influence on projection (VIP) score has been applied to identify primary and specific VOC metabolites for discriminating inoculated and non-inoculated samples through the best-fit model [17], where the compounds having VIP score over 2 and *p* < 0.05 were considered as the discriminating factors [18,19]. The multivariate data analysis was carried out through SIMCA and Metaboanalyst 5.0 (free software).

### 2.6. Clustered Heat Mapping

The heat map of VOC data was drawn via clustering the hyperparameters with respect to storage conditions and type of infection and drawn using Metaboanalyst 5.0. Based on the clustered arrangement of the volatome and their respective colour scale, the prominent VOCs were identified among control, bacteria and fungi-inoculated samples with their parent group.

## 3. Results

### 3.1. VOC Profile of Stored Potatoes

Eleven to twenty-three VOCs were detected in healthy potatoes during storage for three weeks. It can be observed that in the first week of storage, the healthy potatoes had the dominance of aromatic, ester, ketone, sulphur, nitrogen and acid compounds for all storage temperatures (Figure 1A). The most abundant VOC in the healthy potato tubers after the first week of storage was Octadecane, 6-methyl-. However, in the second week, acids, alkane, azide and ester groups were dominant. By the end of three weeks, acids, aldehydes, ketones, alcohol and ester groups had prominence. The GC-MS results of potatoes inoculated with three different microorganisms, *PB*, *AN* and *AF*, revealed various volatile signatures in terms of quantitative and qualitative measures (Figure 1B–D) with respect to storage duration for different storage temperatures. De Lacy Costello et al. [12] detected 3,4-dihydro-2H-pyran, 1-methyl-3-propylbenzene, 2-methyl decane, 2,9-dimethyldecane and 2,9-dimethyldodecane in potatoes inoculated with *Erwinia carotovora*, which is now known as *Pectobacterium carotovorum*; some of these compounds were also detected in the present investigation.

Fungal-inoculated potato tubers easily deteriorated and had early detectable physical symptoms of infection compared to bacterially inoculated potatoes. The volatiles associated with fungal infection were more, and highest (124) in the *AN*-inoculated ones. Very high variability of VOCs was observed within the bacterial and fungal infections, as well as two species of selected fungal strains. However, some volatiles that were common to all selected microbial (bacterial and fungal) infections, including butanoic butyl/ethyl ester, hexanoic ethyl/butyl ester and hexadecane, were not detected in healthy potatoes. Some overall (all storage temperatures) discriminative VOCs belonging to the *PB* culture were 1-butanol and 1-hexanol, which were also detected by Steglinska et al. [20]. Their study revealed several VOC biomarkers, such as sesquiterpenes, dimethyl disulfide, 1,2,4-trimethyl benzene, 2,6,11-trimethyl-dodecane, benzothiazole, 3-octanol and 2-butanol, associated with seed potatoes inoculated with *Fusarium sambucinum*, *Alternaria tenuissima* and *Pectobacterium carotovorum* during a three-month storage period. Some volatiles were common to control potato tubers. n,n-dimethylmethylamine, 1-undecene, acetone and styrene were some of those VOCs, which were also the unique VOCs of sweet potato with soft rot disease caused by *Pectobacterium carotovorum* [21].

For *AN*-inoculated sample, the specific biomarkers observed were alkane, hexadecane, undecane, tetracosane and octadecanoic acid, as well as alkenes such as tridecene and undecene. *AF* inoculated potato tubers had unique volatile signature acids such as hexadecanoic acid and acetic acid derivatives. Some alcohol derivatives were also observed in the *AF*-inoculated sample, including undecanol. These observations suggest a correlation between alterations in the internal metabolism of potato tubers inoculated with the selected microbes. For example, high-carbon-number alkanes were associated with the *AN*-inoculated changes, whereas alcohol groups were associated with the *AF*-inoculated samples, while in bacterial infection, only low-carbon-number alcohol represented a unique VOC profile. Several dissimilar volatiles were explored between the infections, but only some were considered unique VOC signatures.

The study also revealed volatile discrimination between various storage temperatures (4 ± 1 °C, 8 ± 1 °C, 25 ± 1 °C). In addition, significant variability was observed within different storage conditions in inoculated potato tubers compared to the healthy samples, irrespective of the types of infection.

### 3.2. Multi-Variate Data Analysis of VOC Using PCA and PLS-DA

The PCA of the multivariable data of VOCs obtained from potato samples inoculated with *PB, AN* and *AF* and the control group, stored under different storage conditions, is represented in Appendix A, where the samples were assigned to classify through an unsupervised manner into three distinct classes based on the features of the VOCs belonging to specific groups. Hence, the variability among VOCs was determined in an unsupervised manner, and the fitness of the model was evaluated based on R^2^ and Q^2^ values. Accordingly, it was observed that the PCA could not classify and distinguish the samples in space through their distribution characteristics, with lower values of both R^2^ and Q^2^. In the case of the sample stored at 25 °C, the PCA model fitting accounted for 37.4% variance (Appendix A), in the manner of PC1 contributing 19.1% and PC2 contributing 18.3% of the variance. Further, the model fitting obtained R^2^ of 44.7% and Q^2^ of 46.4%. Similarly, in the case of VOCs obtained from samples at 8 °C and 4 °C, their respective PC1 explained 16.7% and 25.2%, respectively, while PC2 explained 13.1% and 17.8% of the variance, with R^2^ values of 46.4% and 42.9% and Q^2^ values of 44.3% and 45.3%, respectively. The VOC clusters of different samples were found to be completely overlapped. However, the extent of overlapping of the clusters was different with respect to the storage conditions.

Further, the data analysis and classification of VOCs were carried out through a supervised model of discriminant features, i.e., partial least squares discriminant analysis (PLS-DA). PLS-DA eliminated the influence of uncontrolled variables, and the model was fitted to the normalized experimental data (Figure 2). Figure 2A shows the graphical representation of the PLS-DA model fit with the score to the non-infected and inoculated potatoes with selected microbial species stored at 25 °C. As per the score plot, the selected two principal components explained 27.2% of the data variance. The PLS-DA model shows the best fit with a higher value of R^2^ (97.6%) and Q^2^ (70.2%) to the VOC data. The position of the points on the graphs represent the respective storage period and a particular type of infection with the samples.

The score plot was drawn to study the classification features among the four distinct groups of the samples. From Figure 2A, it was observed that there was a distinct separation among all three inoculated samples from the control sample without any infection. Among the infections, the *PB* infection of shows a distinct separation from the other two (*AN* and *AF*) fungal infections. A slight overlap was observed between the data clusters of *AN*-inoculated and *AF*-inoculated samples, especially among the points representing elapsed storage time. Interestingly, the points representing control, non-inoculated tubers during the initial storage time (first two weeks) were located in the rectangular part of PC1 > 0 and PC2 < 0. With a further increase in storage time, the position shifted to the rectangle representing PC1 < 0 and PC2 > 0. A similar trend of shifting of points was also observed in the *AN*-inoculated sample, i.e., points representing the first two weeks were present at PC1 < 0 and PC2 > 0, while at week 3, samples were presented at PC1 < 0 and PC2 < 0. The smallest shift of position was observed in the *AF*-inoculated samples among the storage periods, i.e., all points located at PC1 < 0 and PC2 < 0. The highest contribution of PC1 was shown in the non-inoculated control sample, whereas the *PB*-inoculated sample contributed the most to PC2.

The best-fit PLS-DA model was further validated through applying a permutation test (200 permutations) to the data and checking its reliability. As per the test results, the model was found to be reliable with R^2^ = (0.0, 096.7%) and Q^2^ = (0.0, 18.6%) as intercepts [22]. Figure 3A represents the VIP score plot of the VOCs in which the compounds having VIP score > 2 and *p* < 0.05 were the key contributors to the discrimination of control and inoculated samples among themselves based on the PLS-DA model.

Similarly, for the experimental potato samples stored at 8 °C, Figure 2B represents the model fit score plot of the PLS-DA fitted to VOC data recorded from the experimental samples of stored potatoes. It was observed that component 1 of the score plot explained more variance (12.7%) than component 2 (11.7%); both components combined explained about 24.5%. The PLS-DA model best fits the VOC data with an R^2^ value of 97.2% and Q^2^ value of 41.0%. Figure 2B reveals that there was a distinct separation among all four types of samples (control and inoculated) taken for storage. The *AF*-inoculated sample occupied the largest size of the cluster while it was comparatively very small, with the samples stored at 25 °C. The sample inoculated with *PB* showed distinct separation from the other two fungal infections (*AN* and *AF*) with a defined and unique trend of change in VOC with respect to storage time. The considerable change in PC2 values of *AF*-inoculated, *PB*-inoculated and control samples, as well as slight changes in PC1 values of *AN*-inoculated samples, were observed due to the changes in the profile in terms of concentration as well as the type of VOC with respect to the change in the storage period. The shift of points on the score plot represented the change in the respective PC values. Such a shift of points representing VOCs generated at a particular storage time was found to be the highest in *AF*-inoculated samples, followed by *PB*-inoculated and control. Accordingly, the points representing *AF*-inoculated tubers after the first week of storage time were located in the rectangular part of PC1 < 0 and PC2 < 0. After a further increase in storage time, the position is shifted to the rectangle representing PC1 < 0 and PC2 > 0 and again shifted to the rectangle representing PC1 < 0 and PC2 < 0 with a change in PC values. Such a typical type of shift might be due to the change in the concentration of the group of VOCs as well as new secondary metabolites generated during temperature-influenced changes in metabolic processes. A similar but reverse trend of shifting of points was also shown in the control sample, i.e., points representing the first two weeks were present at PC1 > 0 and PC2 < 0 while the weeks 3 sample was at PC1 > 0 and PC2 > 0. The smallest shift of position was observed in the *AN*-inoculated samples among the storage period, i.e., mostly located at PC1 < 0 and PC2 < 0 and slightly shifted to PC1 > 0 and PC2 < 0. The PC-inoculated sample showed the unique trend of a decrease in PC2 value while an increase in PC1 value with respect to an elapsed storage period of up to three weeks. The change in position of the points within the cluster of control and *AN*-inoculated samples is found to be much less than the *AF* and *PB* inoculated samples. On the other hand, the Euclidean distance between the points present in the clusters of *AF* and *PB*-inoculated samples was higher, which refers to the significantly higher rate of change in the VOC profile of inoculated tubers than control and *AN*-inoculated samples over the storage time. The best model was further validated through the 200-permutation-based test in which the model was found to be reliable with an R^2^ intercept of (0.0, 99.1%) and a Q^2^ intercept of (0.0, 38.4%), and both lines show a slope greater than zero [23]. The VOC compounds showing VIP score > 2 and *p* < 0.05 in Figure 3B were the key discriminating VOCs for the PLS-DA-based classification of the selected four types of samples.

Figure 2C represents the PLS-DA model fit utilized for the VOC-based classification of the potato sample with the selected type of infection. The first two PCs of the score plot represent 30.2% of the total variance. The PLS-DA model was best fitted to the VOC data with higher values of R^2^ (96.4%), and Q^2^ (65.4%) shows its best reliability and predictability. Figure 2C shows that there was a distinct separation among all three inoculated samples among themselves as well as with the control. A slight overlap between the clusters of *AN*-inoculated and *AF*-inoculated samples was observed, especially among the points which represented the end of the storage period, i.e., the third week. Overlapping was also observed within the individual cluster among the points representing VOCs generated with respect to storage time. The overlapping might be due to the similar type of VOC emanating through a similar type of biochemical reaction and metabolite formation because of the same class of infection, i.e., fungal species. Here, for samples stored at 4 °C, the *AN*-inoculated sample occupied the largest size of the cluster, while it was comparatively very small when the samples were stored at 8 °C. A similar type of reverse trend was also observed with the *AF*- and *PB*- inoculated samples compared with 8 °C of storage temperature. The increased size of the cluster of *AN*- and *PB*-inoculated samples might be due to the higher number of discriminating VOCs obtained when the sample was stored at 4 °C.

A considerable change in PC1 as well as PC2 values of *AN*-inoculated, *PB*-inoculated and *AF*-inoculated samples, compared to slight changes in PC2 values of the control sample, were observed within the cluster due to changes in the VOC profile with respect to the change in storage time. The shift of points on the score plot represents the change in respective PCA values. Such a shift of points representing VOCs generated at a particular storage time was found to be highest in the *AN*-inoculated sample, followed by the *PB*-inoculated and *AF*-inoculated samples. Therefore, in the cluster representing *AN*-inoculated tubers, the points representing the first week and third week of storage time were located in the rectangular part of PC1 < 0 and PC2 < 0, while they were shifted into the rectangle representing PC1 < 0 and PC2 > 0 during the storage time of 2 weeks. Such an intermediate shift might be due to the formation of temporary metabolites, which may further be degraded into parent types of metabolites. An exactly similar type of shifting of points was also observed in the case of the sample inoculated with *PB*, i.e., points representing the first and third week were present at PC1 > 0 and PC2 > 0, while the second-week sample was present at PC1 > 0 and PC2 < 0. There was no shift of position of the points in the clusters of the *AF*-inoculated sample and control sample. The Euclidean distance between the points present in the clusters of *AN*- and *PB*-inoculated samples was higher, and hence the VOC profile of these samples changed at a significantly higher rate than control and *AF*-inoculated samples over the storage time. The best model was further validated through the 200-permutation-based test in which the model was found to be reliable with an R^2^ intercept of (0.0, 97%) and Q^2^ intercept of (0.0, 25.6%), and both lines show a slope greater than zero [24]. The VOC compounds that show a VIP score > 2 in Figure 3C were the key discriminating VOCs for the PLS-DA-based classification of the selected four types of samples.

### 3.3. Clustered-Heat-Map-Based Prominence of Volatome

Groups of VOC-based heat maps with hierarchical clustering simultaneously applied to both rows and columns were prepared using Ward’s method and are represented in Figure 4. The intermediate values of the concentrations were based on their proximity to the maximum or minimum value. For the generation of the said clustered heat map, the VOC data were arranged in the form of their parent group and a unique number of individual VOCs. Further, the numerical data of the VOCs were normalized, and this allowed for clustering and colour scaling based on the concentration of the particular compound. Further, the group of compounds that appeared in a dark brown colour scale on the heat map were considered the dominant group of VOC for instantaneous storage conditions (type of infection, storage temperature and storage period).

Accordingly, Figure 4A represents the VOC groups’ clustered heat map emanating and mapping from the stored potato samples at 25 °C. Figure 4A reveals that, for the control non-inoculated samples, the VOCs belonging to the groups of alkane, alkene and acids were dominant in the first week. This dominance was found to be shifted towards the groups of ketone, alkene, acids and esters by the third week of storage. The same group repeated with an increase or decrease in their concentration due to changes in metabolic processes, while the existence of the new dominant group might be due to the formation of new metabolites due to changes in metabolic pathways. In a similar way, in the case of potato samples inoculated with *Pectobacterium carotovorum*, acids, alkane, esters and aromatic compounds were mapped as dominant VOC groups at the first week, while at three weeks, the VOCs’ dominance remained for aromatic compounds only.

The dominance of aromatic compounds, acids and esters observed in the first week of storage time shifted to the amide, ether, azide, amine and sulphur groups of compounds in the third week of the storage time when the potato sample was infested with *AF*. Dominant compounds were also found to be changed with a change in species of fungal infection; likewise, for the *AN*-inoculated potato sample, acid, alkane, nitrogen, ester, aromatics and aldehydes were the dominant groups of VOC compounds mapped at one week of storage period, while they shifted to ester, acid, ether, amide and ester after three weeks of storage.

The dominant VOC emanating from the stored samples also varied with respect to the storage temperature. Figure 4B shows the heat map of the VOC groups mapped from the potato samples stored at 8 °C. The heat map shows that, in the control (CW1) sample, acids, aromatic, ester, alkane, alcohol and hydrazine were the dominant groups in the first week of the storage period, while it was reduced into ketone, aromatic and acids when storage time reached up to three weeks. The sample inoculated with *PB* shows aromatic, ester, sulphur, acid and alcohol as dominant VOC groups when mapped at the first week (PBW1) of storage time and gets decomposed due to various metabolic changes in such a way that only acids remained as the dominant VOC group after up to three weeks (PBW3) of storage time. At the same time, the *AN*-inoculated sample after one week (ANW1) of storage time especially emitted VOCs belonging to the groups of aromatic, ketone and esters as dominant groups, and after various metabolic processes, they were shifted to esters, alkane, acids, sulphur and aromatic compounds after up to three weeks (ANW3) of storage time. Acid was the only dominant group of VOCs observed at one week (AFW1) of the storage period of the *AF*-inoculated potato sample, while after up to three weeks (AFW3) of storage time, nitrogen, aromatic, amine, ester and ketone were added with acids as dominant groups of VOCs.

The smallest number of VOCs was recorded and mapped from the potato samples stored at 4 °C, while the dominance of particular types of VOC groups was found to be different with respect to the type of infection as well as the duration of storage. The profile of the VOC groups recorded from potato samples stored at 4 °C is represented in Figure 4C in the form of a clustered heat map. From the heat map, it was revealed that for the control sample, ketone, alcohol, acid, sulphur, amines and aromatic compound groups were most dominant in the first week (CW1) of the storage period and further shifted to the groups of esters, acids, alkene and aromatics at three weeks (CW3) of storage time. In a similar way, in the case of potato samples inoculated with *PB*, acids, alkene, esters and alcohol were mapped as dominant VOC groups at the first week (PBW1), while at up to three weeks (PBW3) of storage time, the compound dominance remained and shifted towards acid, alkane, alkene and nitrogen groups of VOCs. Among the inoculated fungal samples, the *AF*-inoculated sample after one week (AFW1) of storage time shows alcohol only as the dominant VOC group, and due to various metabolic processes after up to three weeks (ANW3), the dominance of VOCs shifted towards acids, alcohol, ketone, alkane, amine, sulphur and aromatic compounds. A highly specific type of group was found in the case of *AN*-inoculated samples: likewise, esters and amines at the first week (ANW1) of storage time and acids with amines at the third week (ANW3) of storage time.

## 4. Discussion

A different VOC profile of the control sample compared to the inoculated samples could be due to the evolution of some metabolites related to plant defence mechanisms or as a response to external stimuli. Moreover, at lower temperature storage (4 °C), the conversion of starch into glucose [24] and the development of aromatic compounds might be the reason for the evolution of aromatic and ester compounds in the first week of storage at 4 °C, which later degraded to ketones, which were prevalent groups after the second week of storage. Even though infection-related volatiles were not found under cold storage of 4 °C and 8 °C, this represented the retardation of microbial metabolism at lower temperatures and condensation of volatiles to complex compounds. There might be some internal shifting of metabolic pathways towards some specific secondary metabolism that depends on enzyme activation under lower temperature storage. In the case of fungal infection, the scenario of the VOC profile was different from that of bacterial infection in terms of quantitative and qualitative parameters. The fungal-infection-linked unique volatiles were seen even at 8 °C in week 3, whereas those compounds could be sensed in week 1 during room temperature storage. This, nevertheless, suggests the activity of fungi at 8 °C. Quantitatively, also, volatiles were not variable throughout storage in fungal-inoculated potato tubers. Fungal-infection-related VOCs could not be detected at 4 °C, which further supports the idea of inhibited metabolism at lower temperatures. The main reason might be the easy and sporadic entry of fungal strains to nearby potatoes under storage and fungal resistance up to 8 °C. Even at 8 °C, fungal strains were able to combat with the potato’s internal metabolic pathways and generation of secondary metabolites that were emitted as volatiles [25].

A clear separation of VOCs of different samples as well as storage conditions was not observed during PCA. The said overlapping of the clusters was due to the unsupervised analytics of the PCA model, which cannot remove the influence of uncontrolled variables completely from the data [23].

For the potatoes stored at 25 °C, in the PLS-DA, a slight overlapping among the clusters of *AN*-inoculated and *AF*-inoculated samples, especially among the points, which represented elapsed storage time, might be due to the similar group of VOCs obtained due to the same class of infection, i.e., fungal species. The considerable change in terms of increase or decrease in the PC2 value of *AN*- and *PB*-inoculated samples and PC1 value of control and *AF*-inoculated samples revealed that changes occurred in the VOC profile during the whole storage period irrespective of the type of infection. The change in the position of the clusters shifted to the rectangle representing PC1 < 0 and PC2 > 0 was attributed to the changes in VOC generation due to alterations in metabolic processes and synthesis of secondary metabolites. Differences in the positions of the points within the individual cluster of the sample with changing storage time were found to be lower in the control sample compared to the clusters of all other inoculated samples, which reflected that the Euclidean distance between the points of the control sample was less than the inoculated samples. Hence, the VOC profile of inoculated tubers changed at a significantly higher rate than the control over the storage time.

The increased size of clusters in the *AF*-inoculated potato stored at 8 °C was higher than at 25 °C, which may be due to the evolution of a higher number of discriminating VOC metabolites at this temperature. At a lower temperature of storage (4 °C and 8 °C), the reducing sugar content increases [26], which might have produced related metabolites. Besides, fungi growth is also substantiated in potatoes stored at 8 °C, leading to postharvest losses as a result of heightened fungal activity and sprouting [27]. On the other hand, at 4 °C, the *AN*-inoculated sample occupied the largest size of the cluster, while it was comparatively very small when the samples were stored at 8 °C. It can be inferred that though a lower storage temperature inhibits bacterial growth, it is still conducive to fungal growth.

## 5. Conclusions

Postharvest storage losses in potatoes are attributed to several bacterial and fungal infections. These infections lead to a change in the metabolite profile of the stored potatoes. These volatile metabolites can be mapped and need to be classified to understand and derive the key fingerprints. The bacterial and fungal infections can be separately identified from the healthy potatoes using PLS-DA classifiers. These VOCs can not only be segregated by infection type but also storage week, indicating aging-related changes at different storage conditions. Different biosynthetic pathways followed by these commodities present a distinct VOC profile at lower temperatures, indicating the slowing down of the basic metabolism of the commodities and the inoculated organisms. The survival of fungi, even at lower storage temperatures, also calls for suitable postharvest treatment of stored commodities such as potatoes. With higher R^2^ and Q^2^, the performance of PLS-DA is better than the PCA. The supervised chemometric classifiers developed during this investigation can be used for fast and accurate distinction of infections in potato storage systems to mitigate suitable disinfection measures, as a decision support system, specifically for the application of fungicides or bactericides during storage at different conditions for preventing further losses.

## Figures and Tables

**Figure 1 foods-12-02083-f001:**
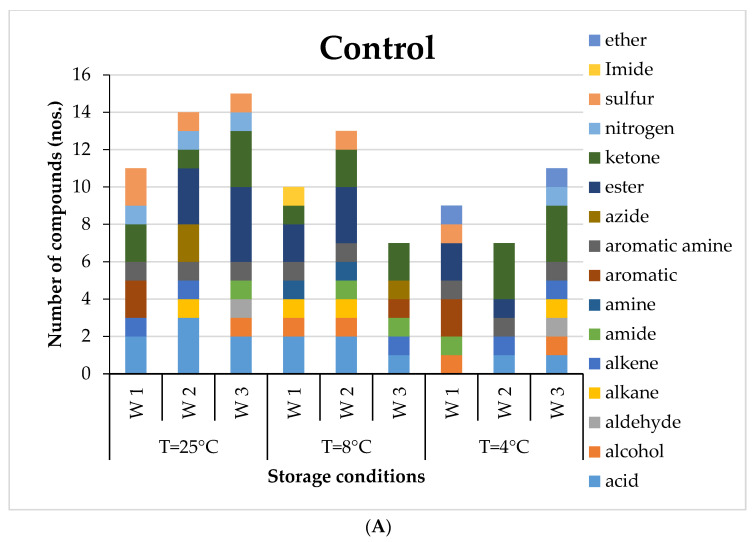
The group of VOCs mapped at specified storage conditions of potato as (**A**) non-infected sample (control), (**B**) *Aspergilus-niger*-inoculated, (**C**) *Aspergilus-flavus*-inoculated and (**D**) *Pectobacterium-carotovorum*-inoculated, for 3 weeks of storage (W1, W2 and W3).

**Figure 2 foods-12-02083-f002:**
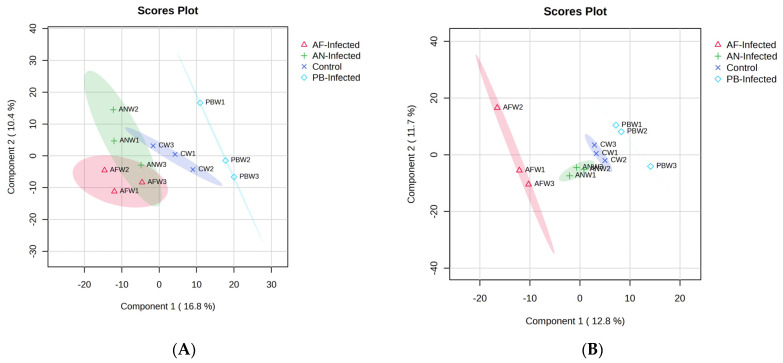
PLS−DA plots of the VOCs recorded from different potato samples stored at (**A**) 25 °C, (**B**) 8 °C and (**C**) 4 °C.

**Figure 3 foods-12-02083-f003:**
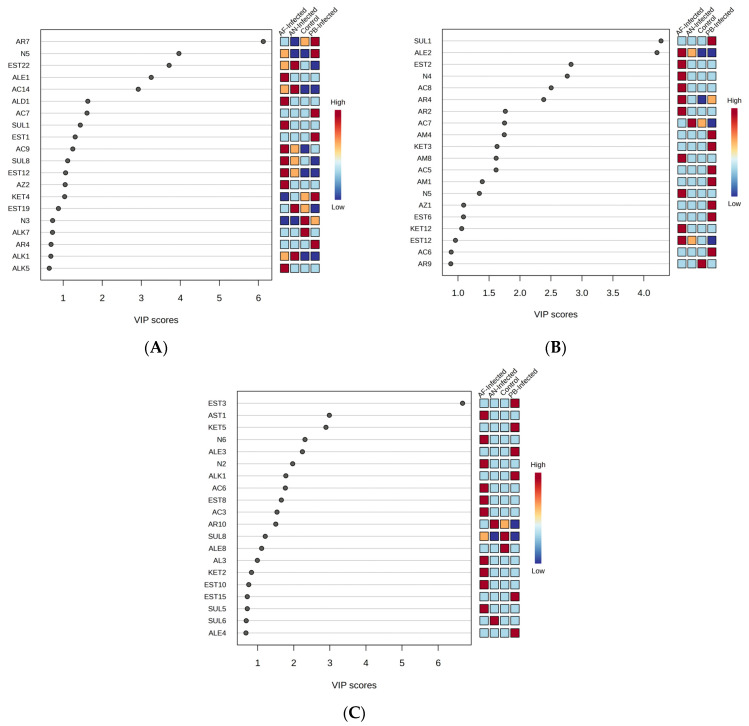
VIP score plot of the discriminating VOCs of PLS-DA of different potato samples stored at (**A**) 25 °C, (**B**) 8 °C and (**C**) 4 °C.

**Figure 4 foods-12-02083-f004:**
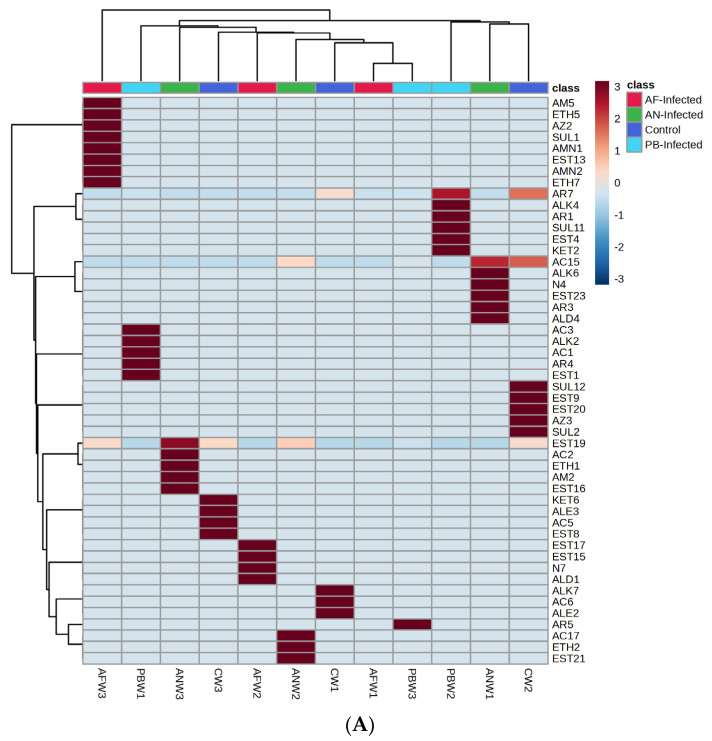
Clustered heat map of group of compounds mapped from different potato samples stored at (**A**) 25 °C, (**B**) 8 °C and (**C**) 4 °C.

## Data Availability

Data is contained within the Appendix A.

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
