# Peer review of "Spotting of Volatile Signatures through GC-MS Analysis of Bacterial and Fungal Infections in Stored Potatoes (Solanum tuberosum L.)"

_foods, 2023, doi:10.3390/foods12102083_

Round 1

Reviewer 1 Report

In this manuscript, the authors qualitatively analyze the changes in volatiles in infected potatoes over a three-week period. The comments are as follows.

1. The result that volatiles differed under different treatments was predictable, but whether volatiles can be used as marker differentiators cannot be determined by the peak area of the substance alone.

2. Is the AN and the ASN in the figures and tables the same treatment, if so, please describe them consistently.

2. Please provide the standard deviation of the mean for the tabular data

3. PCA does not achieve data separation and is not meaningful for data analysis, consider putting this section into supplementary material or selective deletion.

5. Questions about the image format: keep the image size consistent for Figure 1. The substance names in Figure 4 are incomplete, please adjust them.

Author Response

The authors sincerely thank the reviewer for his/ her painstaking effort in reviewing the manuscript, giving constructive criticism and fruitful suggestions. This has improved the quality of the manuscript considerably.

Reviewer 2 Report

The authors did experimentally demanding research on spotting volatile compounds in bacterial and fungal infections of stored potatoes. The paper is very well written and easy to follow. However, I have several technical suggestions and methodological questions:

1. Line 158- you listed “Fig 1 a,b,c” but the text refers to Figure 1 b, c, d. You should also list Figure 1 a in the appropriate part of the text (for example at the end of the sentence starting at line 160). In the text, the letters A, B, C and D should be large as in the picture.

2. Line 166- delete the extra bracket after the 12th reference.

3. In the legend of Figure 1 clarify that W1, W2 and W3 refer to specific weeks of treatment.

4. In general, attention should be paid to the reference to figures in the text. It should be uniform.

5. In Figure 1, the letters for A, B, C and D segments are placed in brackets, and in Figure 2 this is not the case. Uniform that too.

6. The segments A, B, C and D in Figure 1 are not the same size. It should be uniform.

7. In Figure 2 there is the abbreviation ASN which was introduced through the text as AN. It should be uniform. Abbreviations AN, AF and PB should also be clarified in the legend of Figure 2.

8. Line 311- Add “stored at 8 °C”.

9. The second sentence of the second paragraph, page 13 is redundant because its content can be seen in Figure 5, so I suggest that it should be deleted. Figure 5 (C) should instead be cited in brackets as a reference at the end of the first sentence of the same paragraph.

10. I suggest that Supplementary Table should be marked as Table S1.

11. Can you please explain what means “Area percentage of VOCs” in the Supplementary Table? Also, can you tell me which spectra library was used for identifying the VOC and how did you quantify VOC? I think it should be stated in the text.

12. You said that the obtained values are listed in the Supplementary Table as means of 3 replications but you have not listed the standard errors in the Table.

13. I also suggest you should be uniform when you cite the culture Pectobacterium carotovorum spp in the text and Table S1.

Author Response

The authors thank the reviewer for his/her keen observation and suggestions to improve the quality of the manuscript.

Round 2

Reviewer 1 Report

the paper can be accepted in current form.

Author Response

Comments and Suggestions for Authors: the paper can be accepted in its current form. 

Response: The authors thank the reviewer sincerely for his/her positive evaluation of the paper. 

Reviewer 2 Report

Dear authors,

Thank you for considering my suggestions and answering my questions. In my opinion, it is still necessary to make some corrections.

1. Please explain what the dashes in Table S1 mean (not detected or something else).

2. You should be careful how you cite the figures in the text (is it Figure, Fig. or fig.).

3. References should be cited at the end of the sentence before the full stop.

Author Response

Comments and Suggestions for Authors

Dear authors,

Thank you for considering my suggestions and answering my questions. In my opinion, it is still necessary to make some corrections.

Response: thank you very much for the favourable and constructive criticism , which has allowed us to improve this work.

  1. Please explain what the dashes in Table S1 mean (not detected or something else). Response: dash means not detected, a footnote for the same is being included in table S1
  2. You should be careful how you cite the figures in the text (is it Figure, Fig. or fig.). Response: error regretted. Now we have maintained uniformity. Changes are marked in red.
  3. References should be cited at the end of the sentence before the full stop. Response: authors feel that the citations should indicate relevance; if we have to put all the references at the end of the sentence, it may be confusing. For example, we have used one reference for fungi culture and another for bacterial culture, so they should be cited at the respective places. moreover, in the discussion, we cited different works and deduced our own inferences. if we have to put all references at the end, it won't make sense, or it will look like we have quoted someone else's work. we have also confirmed this with earlier papers published in foods. This part may please be considered.